https://doi.org/10.1038/s41467-021-26283-y　　**OPEN**

# A hierarchical cellular structural model to unravel the universal power-law rheological behavior of living cells

Jiu-Tao Hang[1], Yu Kang [2], Guang-Kui Xu [1✉] & Huajian Gao [3,4✉]

Living cells are a complex soft material with fascinating mechanical properties. A striking feature is that, regardless of their types or states, cells exhibit a universal power-law rheological behavior which to this date still has not been captured by a single theoretical model. Here, we propose a cellular structural model that accounts for the essential mechanical responses of cell membrane, cytoplasm and cytoskeleton. We demonstrate that this model can naturally reproduce the universal power-law characteristics of cell rheology, as well as how its power-law exponent is related to cellular stiffness. More importantly, the power-law exponent can be quantitatively tuned in the range of 0.1 ~ 0.5, as found in most types of cells, by varying the stiffness or architecture of the cytoskeleton. Based on the structural characteristics, we further develop a self-similar hierarchical model that can spontaneously capture the power-law characteristics of creep compliance over time and complex modulus over frequency. The present model suggests that mechanical responses of cells may depend primarily on their generic architectural mechanism, rather than specific molecular properties.

[1] Laboratory for Multiscale Mechanics and Medical Science, Department of Engineering Mechanics, SVL, School of Aerospace Engineering, Xi'an Jiaotong University, 710049 Xi'an, China. [2] College of Pharmaceutical Sciences, Zhejiang University, 310058 Hangzhou, China. [3] School of Mechanical and Aerospace Engineering, College of Engineering, Nanyang Technological University, 639798 Singapore, Singapore. [4] Institute of High Performance Computing, A*STAR, 138632 Singapore, Singapore. ✉email: guangkuixu@mail.xjtu.edu.cn; huajian.gao@ntu.edu.sg

The mechanical properties of living cells are of crucial significance for a number of biological processes, such as development, tumor metastasis, and lesions screening[1-4]. Living cells are a complex active material with both solid-like elastic and fluid-like viscous properties. In response to dynamical forces, cells exhibit viscoelastic behavior such as creep and stress relaxation. A striking feature is that, regardless of cell types or cell states (e.g., drug-induced), the complex moduli $E^*$ of cells show a power-law dependence on loading frequency $f$, $E^* \sim (if)^\alpha$, rather than a classical exponential-type response[4-9]. This is similar to experimental observation that, in response to a step force, cell deformation follows a power-law dependence on time $t$, $d \sim t^\alpha$[10-12]. The power-law exponent $\alpha = 0$ or $\alpha = 1$ is indicative of a purely elastic solid or viscous fluid, respectively, and therefore it can reflect the viscoelastic characteristics of the cell. In reality, $\alpha$ usually falls in the range of 0.1–0.5 for different cell types or states[10-14]. Amazingly, the power-law exponent can be collapsed into a universal master curve, which decreases linearly with increasing cell stiffness in a semi-logarithmic plot[14]. In addition, cells exhibit stress stiffening behavior under static loads, in that their instantaneous stiffness increases linearly with the external or internal stresses[15-17].

A variety of mechanical models have been developed to understand the fascinating power-law rheological properties of cells. Andreas et al. used a traditional viscoelastic model consisting of two springs and two dashpots to fit the creep compliance of cells[18]. However, the drawbacks of such linear viscoelastic models are that a large number of springs and dashpots are required to fit the power-law characteristics, often without a clear physical interpretation[13,19]. Fabry et al.[8] succeeded in explaining the power-law behavior by using a soft glassy rheology (SGR) theory, but the predictions based on the SGR theory showed a stress softening behavior, which is opposite to that of living cells. While a tensegrity model described the stress stiffening of cells[15,20,21] and predicted the power-law rheological phenomenon by using viscoelastic tendons[20], it

typically cannot quantitatively tune the power-law exponent and cannot explain the unified relationship between the power-law exponent and cell stiffness. Some bottom–up models, such as those based on polymer molecules, may realize both stress stiffening and power-law rheology, yet the power-law exponent of these models ranges from 0.5 to 0.75[22-26]. To date, to the best of our knowledge, no single model can capture the full phenomenology of the cellular mechanical behavior.

In addition to cells, the power-law rheological behaviors have also been observed in a wide variety of biological materials of different molecular compositions[9,27], suggesting that the common rheological features may have no specific molecular properties but instead arise as a result of similar structural architecture. A successful comprehensive model of cell mechanics is thus likely of a generic and mechanistic character. Here we propose a cellular structural model, which includes membrane, cytoskeleton, and cytoplasm as its main components. By examining its mechanical responses under both step and cyclic loads, we show that this structural model is capable of reproducing the power-law rheology and the unified relation between the power-law exponent and cell stiffness. Subsequently, a self-similar hierarchical theory is proposed to capture the power-law rheology characteristics with tunable power-law exponents in the range of 0.1–0.5.

## Results and discussion

**Model: cell structure.** A finite element-based cell model consisting of membrane, cytoskeleton and cytoplasm is established and then used to simulate the viscoelastic responses of the cell between two microplates, as in experiments[12,13]. In the cytoskeleton, microtubules (MTs) emanate from the centroid, grow straight, and partly reach the membrane[28]. As the MT length varies greatly, there exist some short MTs that do not reach the membrane[29]. Here we use emanative MTs with different lengths as the cytoskeleton (see inset of Fig. 1a). The cell volume is taken as ~3000 μm³, with a diameter of 15 μm and a height of 15 μm[30]. MTs are hollow, tubular structures with an internal radius of

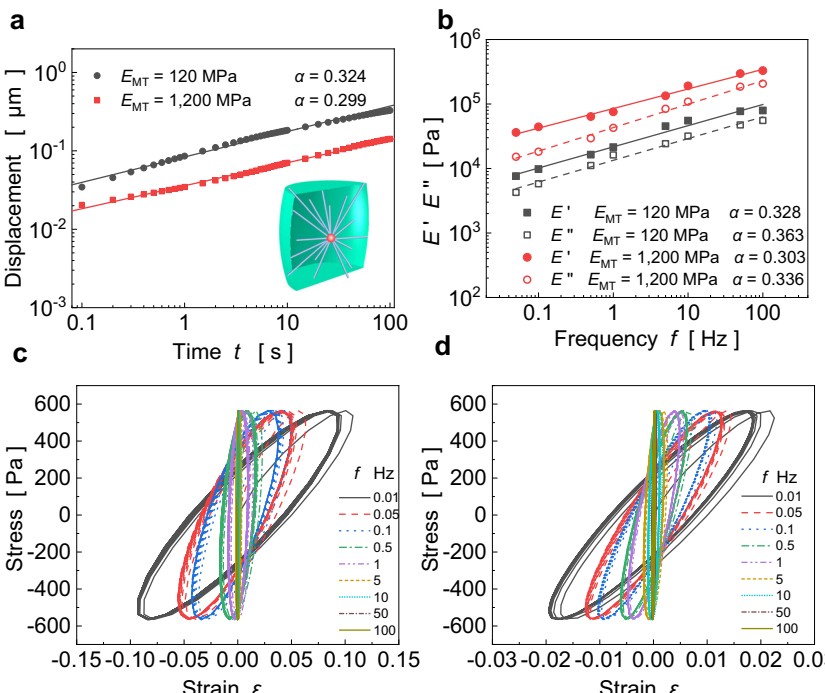

**Fig. 1 Rheological responses of the adopted cell model under step and cyclic loads. a** Displacement over time on the log–log scale for $E_{MT} = 120$ MPa and $E_{MT} = 1200$ MPa. **b** Storage modulus $E'$ and loss modulus $E''$ over the loading frequency $f$ on the log–log scale for $E_{MT} = 120$ MPa and $E_{MT} = 1200$ MPa. Lissajous figures of stress–strain curves under different frequencies for **c** $E_{MT} = 120$ MPa and **d** $E_{MT} = 1200$ MPa, respectively.

7.5 nm and an external radius of 12 nm[31]. The cell membrane is modeled as a one-layer viscoelastic shell with a thickness of 6 nm[32].

**Model: material properties**. In our finite element simulations, the MTs are treated as linear elastic materials and the cytoplasm and cell membrane as Kelvin–Voigt viscoelastic materials. The cytoplasm is a crowded aqueous solution filled with ions and proteins. Hence, different cells exhibit different viscosities depending on the volume fraction of each component in the cytoplasm, as well as the interaction between the cytoplasm and the cytoskeleton. In this sense, the viscous coefficient $\eta$ represents the effective viscosity of the entire cytoplasm. Due to Poisson's effect (the Poisson's ratio of the cytoplasm is 0.37[33]), the transverse elastic expansion of the cytoplasm makes the creep compliance in three-dimension (3D) different from that of the Kelvin–Voigt model (see Supplementary Note 1). The material parameters of each component are listed in Table 1. Here we use the above simple constitutive relation to examine the viscoelastic responses of our model with proposed structural characteristics.

**Power-law rheology**. We first simulate the creep responses of a cell under step loads $\sigma(t) = \sigma_0 u(t)$ with $\sigma_0$ being the amplitude and $u(t)$ the step function, as well as under cyclic loads $\sigma(t) = \sigma_0 \sin(2\pi f t)$ with $f$ being the frequency. Under a step force of 10 nN, Fig. 1a shows that the deformation increases linearly with time on the log–log scale (i.e., $d \sim t^\alpha$), as found in many experiments[10–12], for different moduli of MTs. For cyclic loadings, Fig. 1b shows that both the storage modulus $E'$ and the loss modulus $E''$ exhibit a power-law dependence on the loading frequency $f$, in agreement with experiments[4,5,8,9]. Lissajous figures plot the corresponding stress–strain curves under different frequencies (Fig. 1c, d), and their shapes mimic symmetric ellipses as in experiments[34,35]. Interestingly, the power-law exponent $\alpha$ of the storage modulus obtained in the frequency domain (Fig. 1b) is very similar to that of the creep response obtained in the time

domain (Fig. 1a). Therefore, we use the creep response to investigate the rheological behavior of cells in the sequel. In addition, the power-law exponent $\alpha$ of the loss modulus is slightly larger than that of the storage modulus (see Fig. 2b), which is also observed in experiments[5,8], and will be clarified in a later section.

**Power-law exponent**. For different cell types or cell states, cell stiffness can vary greatly, and the viscoelastic characteristics may differ significantly. By varying the number of MTs, we will show that the cellular stiffness can be regulated, and their creep responses are analyzed to assess the generality of this model. For example, we plot the calculated creep responses of cells with respect to the number of emanative MTs (Fig. 2a). It can be seen that increase in the amount of MTs can reduce the power-law exponent from 0.564 to 0.189. These findings demonstrate that the cell behaves more like a solid as the number of MTs increases and more like a liquid when it decreases. In fact, changes in MT number and stiffness are among a number of factors that can alter the power-law exponent of cells. Similarly, changes in mechanical properties of other components of the cytoskeleton (microfilaments (MFs)[5,6] and intermediate filaments (IFs)[36,37]) or the cytoplasm[38,39] can also regulate the power-law exponent of cells. Therefore, it is possible that, through re-configuring the network of the cytoskeleton or changing the mechanical properties of the cytoplasm, the power-law exponent can be quantitatively tuned in the range of 0.1–0.5, which may explain why the power-law exponent differs for different cell types or states (e.g., drug-induced)[5,7,8]. Besides, $\alpha$ is found to decrease linearly with increasing cellular stiffness $E_0$ in a semi-logarithmic plot (Fig. 2b), which will be further investigated shortly. Hence, our model can yield a wide range of values for the power-law exponent $\alpha$, which cannot be achieved by the existing SGR theory or bottom–up models.

**A self-similar hierarchical model**. In cells, abundant MTs and MFs interpenetrate with each other to form a 3D cytoskeleton network bathed in the cytoplasm[40–42] composed of water, solutes, and small molecules. Based on the structural characteristics of the cells, we propose a self-similar hierarchical model to study their rheological behavior. Since the cytoplasm is ubiquitous, its spatial component can be discretized and regarded as infinite springs with elastic stiffness $E_1$ immersed in a viscous fluid with the viscous coefficient $\eta$, as shown in Fig. 3a. The cytoplasm is treated as the 1st-level hierarchy, which fills the entire cell. A single MT can be discretized into an infinite series of springs with elastic stiffness $E_2$, with each node connected to the cytoplasm (the 1st-level structure). Then, each MT embedded into the cytoplasm is considered as the 2nd-level self-similar hierarchy with the

**Table 1 Material properties of the cellular components.**

|  | Cytoplasm | Membrane | Microtubules |
|---|---|---|---|
| Elastic modulus (Pa) | 100[32] | 1000[32] | $1.2 \times 10^9$ [31] |
| Poisson's ratio | 0.37[33] | 0.3[32] | 0.3[31] |
| $\tau$ (s) | 300[a] | 20 | 0 |

[a]At present, there is no clear data on the viscosity $\eta$ of cytoplasm. Based on experimental data[12], we take $\tau = \eta/E$ as 300 s.

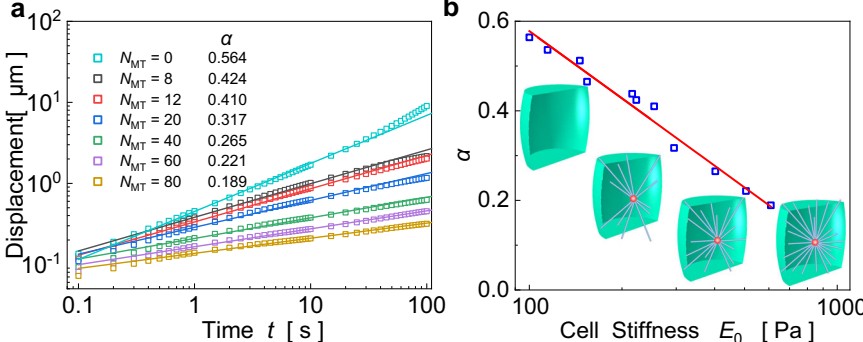

**Fig. 2 Simulated power-law rheology of cells with adjustable power-law exponents as the number of emanative MTs is varied. a** Displacement responses of cells with different numbers of emanative MTs (from 0 to 80). **b** The power-law exponent decreases linearly with the increase of cellular stiffness $E_0$ in a semi-logarithmic plot.

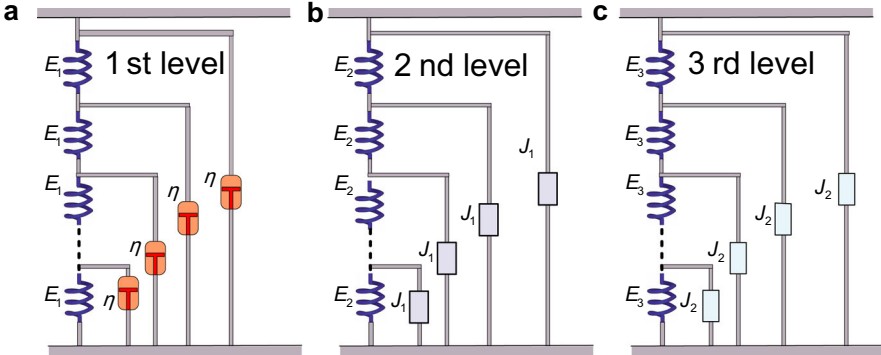

**Fig. 3 A self-similar hierarchical rheological model of the cells. a** The 1st-level hierarchy consists of a ladder-like structure with springs along the struts and dashpots on the rungs of the ladder. **b** The 2nd-level self-similar hierarchy is constructed with the first-level hierarchy represented as a building block $J_1$. **c** The 3rd-level self-similar hierarchy is constructed with the second-level hierarchy represented as a building block $J_2$.

1st-level hierarchy as a building block, as shown in Fig. 3b. Because there are many MTs with different lengths and orientations in the cell, the whole structure network is equivalent to a large number of parallel MTs (the 2nd-level hierarchy) connected by springs (the transverse expansion of the cytoskeleton and the cytoplasm) with elastic stiffness $E_3$. Then the entire cell can be modeled as the 3rd-level self-similar hierarchy with the 2nd-level hierarchy as a building block (see Fig. 3c). In this way, from a macroscopic perspective, the cell is treated as a three-level self-similar hierarchical structure with $E_1$, $E_2$, and $E_3$ representing, respectively, the effective stiffness of the cytoplasm, MTs in the load direction, and the transverse expansion of the cytoskeleton and the cytoplasm and η representing the effective viscosity of the entire cytoplasm.

Because of the orderly arrangement of springs and dashpots in the model, we propose a simple yet robust method to obtain the creep compliance and complex modulus of this self-similar hierarchical model. The creep compliances of 1st-, 2nd-, and 3rd-level hierarchies over time are (see Supplementary Note 2)

$$J_1 = (-1 + \sqrt{1 + 4t/\tau})/2E_1, \tag{1}$$

$$J_2 = (-1 + \sqrt{1 + 4E_2 J_1})/2E_2, \tag{2}$$

$$J_3 = (-1 + \sqrt{1 + 4E_3 J_2})/2E_3. \tag{3}$$

respectively, where $\tau = \eta/E_1$. When $t \gg \tau$, the creep compliances become $J_1 \sim t^{0.5}$, $J_2 \sim t^{0.25}$, and $J_3 \sim t^{0.125}$, suggesting the scale-free power-law rheology. As shown in Supplementary Note 2, the power-law exponents $\alpha$ of the 2nd- and 3rd-level hierarchies fall in the ranges of 0.25–0.5 (Supplementary Fig. 3a) and 0.125–0.5 (Supplementary Fig. 3b), respectively. Interestingly, the upper limit of $\alpha$ in our self-similar hierarchical model is 0.5, which is found and not understood in experiments[11,14]. In the case of $\alpha = 0.5$, the cell architecture may be messy, but the 1st-level hierarchical structure (i.e., the cytoplasm) serves as the major constituent bearing the external force with creep response $J_1 \sim t^{0.5}$. By varying the parameters (e.g., $E_1$, $E_2$, $E_3$, $\eta$), the power-law exponent can be tuned within the range of 0.1–0.5 (see Supplementary Fig. 3), as found in many experiments[10,12,13]. Thus, the present model can spontaneously capture the power-law characteristics of cell rheology. More importantly, one can quantitatively regulate the power-law exponent by varying the relevant stiffness or viscosity of cells, with these properties relating to cell types or states.

The complex moduli of 1st-, 2nd-, and 3rd-level hierarchies over frequency are given by (see Supplementary Note 3)

$$G_1 = E_1 \frac{1 + \sqrt{1 + 4(i\omega\tau)^{-1}}}{2(i\omega\tau)^{-1}}, \tag{4}$$

$$G_2 = \frac{G_1 + \sqrt{G_1^2 + 4E_2 G_1}}{2}, \tag{5}$$

$$G_3 = \frac{G_2 + \sqrt{G_2^2 + 4E_3 G_2}}{2}. \tag{6}$$

When the frequency is very low (e.g., $\omega\tau \ll 1$), the complex moduli can be rewritten as $G_1 \sim (i\omega\tau)^{0.5}$, $G_2 \sim (i\omega\tau)^{0.25}$, and $G_3 \sim (i\omega\tau)^{0.125}$. The power-law exponents of the 2nd- and 3rd-level hierarchies are, respectively, in the ranges of 0.25–0.5 and 0.125–0.5, which are consistent with those of creep compliances over time. With increasing frequency, the loss modulus increases faster than the storage modulus, as the term $i\omega\tau$ increases the proportion of the imaginary part of the complex modulus (see Eq. (4)). Thus, the power-law exponent of the loss modulus will be larger than that of the storage modulus at high frequencies, as in experiments[5,8] and our simulation results (Fig. 1b).

Using this self-similar hierarchical model, we have analyzed experimental data on the complex modulus over frequency[8]. The predictions of the present model are in excellent agreement with experimental results on both storage and loss moduli for HASM cells under different drug treatments, as shown in Fig. 4. The drug Histamine[43] can enhance the permeability of cells, which reduces the cytoplasmic stiffness $E_1$ (see Fig. 4b). This drug also promotes cell contraction that can increase the stiffness of the cytoskeleton network ($E_2$ and $E_3$). For cells treated with DBcAMP, the contraction of cells is inhibited[8,44], which reduces the stiffness of the cytoskeletal network ($E_2$ and $E_3$), as shown in Fig. 4c. When the cells are treated with cytochalasin D[8], the cytoskeleton is dissolved, resulting in a reduction in stiffness ($E_2$ and $E_3$) of the cytoskeletal network (see Fig. 4d). Furthermore, the fitted values of the cytoplasmic viscosity for HASM cells under different drug treatments are consistently close to the measured value (1.41 Pa · s) from experiments[8]. In addition, the self-similar hierarchical model can also be used to study the power-law rheology observed in the cytoplasm whose storage modulus follows a similar power-law form $G' \sim \omega^\beta$ with $\beta = 0.15$[38]. When using this model to investigate the rheological response of the cytoplasm, the structural details of the cytoplasm should be considered. The interstitial fluid inside the cytoplasm (containing water, ions, and small proteins) can be treated as the 1st-level hierarchy, the large-scale proteins in the cytoplasm

as the second-level hierarchy, and the interactions between the proteins as the third-level hierarchy. In this sense, the present model can be extended to investigate the dynamical mechanical response of the cytoplasm. With this self-similar hierarchical model, one can describe, explain, and predict the rheological behavior of living cells with different types or states, as well as the viscoelastic cytoplasm. The application of this model can also avoid the time-consuming computation of a large number of finite element models.

**The relation between power-law exponent and cellular stiffness.** As shown in our simulation results (Fig. 2b) and summarized from experimental data[14], the power-law exponent decreases linearly with the cellular stiffness in a semi-logarithmic plot. To explore the underlying mechanisms, we calculate the creep responses of cells (Fig. 5a) and plot the power-law exponent with respect to the cell stiffness in Fig. 5b. It can be seen from Fig. 5a that, when the stiffness is not high, the creep compliance can intersect at a point $(\tau_0, j_0)$ and be described as $J(t) = j_0(t/\tau_0)^{\alpha}$, where $j_0$ is a characteristic prefactor and the time $t$ is normalized by a timescale $\tau_0$. Letting $J_0 = j_0/\tau_0^{\alpha}$ denote the

compliance value at $t = 1$ s, we get

$$\alpha = -\frac{\log(1/J_0)}{\log(1/\tau_0)} + \frac{\log(j_0)}{\log(\tau_0)}. \tag{7}$$

From the above, the power-law exponent $\alpha$ is found to decrease linearly with the cell stiffness $(1/J_0)$ in a semi-logarithmic plot. Furthermore, by analyzing the data in Fig. 5b, we obtain the value of $\log(j_0)/\log(\tau_0) = 0.55$, which is almost the upper bound of power-law exponents observed in both experiments[14] and our model. Kollmannsberger et al.[11] experimentally presented that $j_0$ and $\tau_0$ are approximately $5.59 \times 10^{-7}$ Pa$^{-1}$ and $5.5 \times 10^{-13}$ s, respectively. Our predicted value 0.55 is also close to their experimental data (0.51). It should be noted that, with the increase of cell stiffness, the creep response curves no longer pass through the intersection point, rather they become parallel with each other. In this case, the relation in Eq. (7) no longer holds, and the power-law exponent (the slope of the curve) tends to be a constant, as shown in Fig. 5b. In our self-similar hierarchical model, the power-law exponent of the third-level hierarchy takes the lower limit of 0.125, when the cell stiffness is extremely high.

Here we summarize the existing experimental results[5,11,45,46] for different cell types and states and plot the power-law exponent with respect to the cellular stiffness, as shown in Fig. 6. It is clearly seen that our predictions agree well with the experimental results

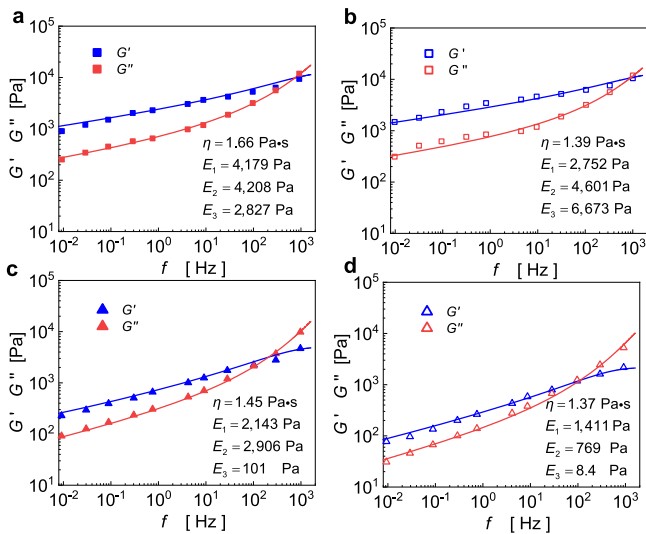

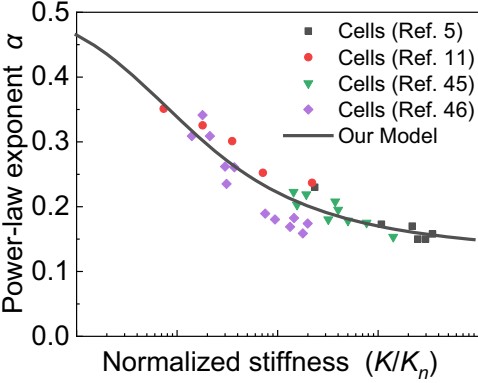

**Fig. 6 The relation between power-law exponent and normalized cellular stiffness.** The power-law exponent $\alpha$ versus cellular stiffness of different cell types and states in experiments[5,11,45,46] collapse onto a master curve. The stiffness of the cell corresponds to the inverse of the creep compliance $J(t)$ measured at $t = 1$ s or the storage modulus $G'(\omega)$ measured at an angular frequency $\omega = 1$ Hz.

**Fig. 4 The self-similar hierarchical model fits well with experimental data from ref. [8] on both storage and loss moduli under different drug treatments. a** Untreated and treated with **b** histamine, **c** DBcAMP, and **d** cytochalasin D.

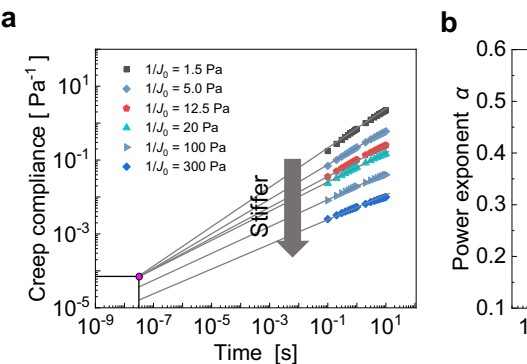

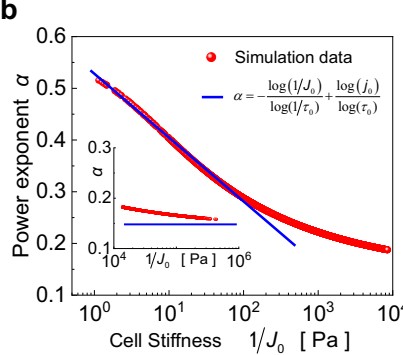

**Fig. 5 Relation between power-law exponent and cellular stiffness. a** The creep responses of cells with different cell stiffness intersect at one point at relatively low stiffness and become parallel at sufficiently high stiffness. **b** The power-law exponent $\alpha$ versus cellular stiffness $1/J_0$. Inset shows the power-law exponent when the cell stiffness is high.

and the cells become more solid-like as their stiffness increases. These results confirm our predictions that, for moderate cellular stiffness, the power-law exponent decrease linearly with the cell stiffness in a semi-logarithmic plot. Moreover, the power-law exponent of cells gradually converges to a certain threshold with increasing stiffness, which was not discussed in previous literature[14]. These broad agreements between experimental findings and our predictions show the robustness of our self-similar hierarchical model in describing cell rheology.

In summary, we have proposed a cellular structural model that successfully captures the power-law rheological characteristics of cells where the power-law exponent can be subtly tuned by the stiffness or the architecture of cytoskeleton. Based on the structural features of this model, we further develop a self-similar hierarchical model of cells to describe their power-law rheological behavior of creep compliance over time and complex modulus over frequency. The predictions of our model are in broad agreement with a vast variety of experiments involving different cell types or cell states. When studying the creep response of cells under small deformations, we have ignored the effect of IFs, since they contribute little to the cortical stiffness in this case[47]. Very recently, Hu et al. studied the effect of IFs on the mechanical properties of cells and showed that, under large deformations, the IF network behave as a strain-stiffening hyperelastic network that substantially enhance the strength, stretchability, resilience, and toughness of cells[36]. Supplementary Note 4 shows that, by treating IFs and MFs as strings in a prismatic tensegrity structure, the cells can exhibit the remarkable strain-stiffening behavior found in experiments,[15–17,36] while holding the rheological characteristics. In addition, IFs play an important role in the mechanics of epithelial monolayers[37,48], which can also be studied by our model. This suggests a strong potential of self-similar hierarchical models for investigating the mechanics of natural biological materials.

## Methods

We adopt a cylinder bulging out in the middle as a general geometric representation of a single cell taken from ref. [12]. The geometrical and mechanical properties of each part are given in the text. The commercial finite-element software Abaqus 6.13-1 and Python scripts are used to generate the model configurations and evaluate the finite-element model solutions. The cytoplasm was meshed with eight-noded linear brick and hybrid elements. The membrane was meshed with four-noded shell reduced integration elements. The MT was meshed with two-noded linear beam elements in space. The embedded constraint elements between the cytoskeleton elements and the "host" cytoplasm elements are used to confine the translational degrees of freedom of embedded nodes. A tied constraint between the membrane and the cytoplasm is used to ensure the displacement continuity.

We performed two types of loading conditions. The first was a step loading condition aimed to characterize the creep response of the cell in the time domain. The power-law exponent is calculated as the slope of the creep compliance over time on the log–log coordinate. The second was a cyclic loading condition aimed to characterize the complex modulus of the cell in the frequency domain. The power-law exponent is calculated as the slope of the complex modulus over frequency on the log–log coordinate. For both types of loading conditions, we used the same geometric and material model and obtained similar power-law exponents.

For both step and cyclic loads, we simulated the viscoelastic cytoplasm and viscoelastic membrane by the Kelvin–Voigt model with the following constitutive relations:

$$\sigma_{cyto} = E_{cyto}\varepsilon_{cyto} + E_{cyto}\tau_{cyto}\dot{\varepsilon}_{cyto}, \tag{8}$$

$$\sigma_{mem} = E_{mem}\varepsilon_{mem} + E_{mem}\tau_{mem}\dot{\varepsilon}_{mem}, \tag{9}$$

where $E_{cyto}$ and $E_{mem}$ represent the elastic moduli of the cytoplasm and membrane and $\tau_{cyto}$ and $\tau_{mem}$ the relaxation times of the cytoplasm and membrane, respectively. For MTs and MFs, we adopt linear elastic constitutive relations with corresponding elastic moduli $E_{MT}$ and $E_{MF}$. The relevant parameters were taken as: $E_{cyto} = 100\,Pa$, $\tau_{cyto} = 300\,s$, $E_{mem} = 1000\,Pa$, $\tau_{mem} = 20\,s$, $E_{MT} = 1200\,MPa$, and $E_{MF} = 2400\,MPa$. A detailed description of the Kelvin–Voigt model can be found in Supplementary Note 1. The detailed geometric parameters of the cell structure can be seen in the "Model: cell structure" section. All simulations were carried out by using the commercial finite element software Abaqus 6.13-1 and can be set

automatically by running a python script (see Supplementary Software 1) in Abaqus 6.13-1.

## Data availability

The authors declare that the data supporting the findings of this study are available in the Source data file provided with this paper. Any data can be made further available upon reasonable request. Source data are provided with this paper.

## Code availability

The python scripts code is included in the Supplementary Software 1 file.

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

## Acknowledgements

Financial supports from the National Natural Science Foundation of China (Grant No. 12072252) and the Natural Science Basic Research Plan in Shaanxi Province of China (Grant No. 2019JC-02) are acknowledged.

## Author contributions

G.-K.X. and H.G. designed the research. J.-T.H. and G.-K.X. performed the research. J.-T.H., Y.K., G.-K.X., and H.G. analyzed the data and wrote the article.

## Competing interests

The authors declare no competing interests.
