## [Peer Review File · Nature Communications]

A hierarchical cellular structural model to unravel the universal power-law rheological behavior of living cellsREVIEWER COMMENTS

Reviewer #2 (Remarks to the Author):

In this manuscript Hang et al. introduce a theoretical model which accounts for the essential mechanical properties of cell membrane, cytoplasm and cytoskeleton. They demonstrate that their model can correctly reproduce not only the universal power-law characteristics of cell rheology but also the relationship between the power-law exponent and cell stiffness. In fact, the power-law exponent is found in this study to be in the range between 0.1 and 0.5, as observed in many experiments, and decreases linearly with the logarithm of cell stiffness. To the best of my knowledge, these results together can not be captured by any existing model of cell mechanics. The Authors also develop a self-similar hierarchical model that can reproduce the power-law dependences of the creep compliance with time and of the storage and loss moduli with frequency. Taken together, these method developments are noteworthy and will contribute significantly to the progress in cell mechanics.

The conclusions of this work are clearly supported by the presented results. I do not find any flaws in the data analysis, interpretation or conclusions. The methodology is sound. The only weakness of this manuscript is that there may be not enough detail provided in the methods for the work to be reproduced. Since the Authors use commercial finite-element software (Abaqus), they could simply deposit exemplary input/output files alongside with Supplementary Notes. It also would be a fine service to the community if the Authors could deposit the Python scripts used in this work to generate the cellular configurations and evaluate the finite-element model solutions.

Suggestion: The predictions of the new model seem to be in quantitative agreement with a vast variety of experiments involving different cell types and states. I am wondering if it would be useful to present (in a new figure 1) the universal master curve of reference 14 (Annu. Rev. Mater. Res. 41, 75-97, 2011) and contrast it with the simulation results reported here.

Reviewer #3 (Remarks to the Author):

This manuscript described a theoretical framework to quantitatively understand the widely observe powerlaw rheology behavior of living cells. To my knowledge, this has not been achieved previously in a way as quantitatively as this manuscript. This model thus would be very valuable in understanding cells and their mechanical behaviors under different conditions. Thus I think this manuscript met the standard of Nature Communications. Nevertheless, I have some comments that should be addressed prior to publication.

1. Power-law power is adjusted by the amount of MT as MT is an elastic component here. This suggests different power in experiments are from a different MT content which is not necessarily the case. I do understand the difficulty of the modeling that it is unnecessary to include more diverse components. But the authors should discuss this result, especially what it indicates to avoid confusion or misinterpretation.

2. Is ita on page 8 the same viscosity as the one used in Table 1, as they have the same symbol? If this is the background fluid, this typically is thought to be a few times more viscous than water according to the diffusion test, as it is an aqueous solution with proteins in it. The viscosity in Table 1 should be more like an effective viscosity of the entire cytoplasm. Otherwise, a timescale of 300 s won't make sense. Or this viscosity here is not the viscosity of the background fluid but rather an effective viscosity of the entire cytoplasm? This should be clarified.

3. Comparing Fig 4a (untreated) and 4d (cytoD treated), why are E2 and E3 change upon cytochalasin D treatment? CytoD dissolves F-actin but E2 is the stiffness of microtubules. Discussion regarding this would be helpful.

4. While this model successfully describes the power-law rheology measured on the level of the whole cell and the cortex, the model itself doesn't separate the thick F-actin rich cortex and the

more diluted internal cytoplasm. Nevertheless, the cytoplasm also shows the classical power-law rheology as shown by many groups (for example <https://doi.org/10.1016/j.cell.2014.06.051>), with a power ranging from 0.1 to 0.5 as well. It seems this model can also do a great job in describing the power-law rheology observed in the cytoplasm alone. As the cytoplasm is known to be much softer than the whole cell or the cell cortex, this distinction would be important to acknowledge. It would be great if some discussion can be added.

5. On page 7, it is stated that MTs and microfilaments cross and connect with each other. They indeed interpenetrate with each other but not necessarily connect to each other.

6. It is known that the cell cytoskeleton is composed of F-actin, MTs and also intermediate filaments. Recently, the role of cytoskeletal intermediate filaments in cell mechanics has become more clear. For example, <https://doi.org/10.1073/pnas.1903890116>. Given each component may have a different mechanical contribution to the overall mechanics, can the authors add discussion on this?

Response to Reviewers

The authors wish to thank the reviewers for their very helpful comments and suggestions. The paper has been revised carefully, and the main changes are marked in blue. Below is an itemized response to each reviewer's comments.

Reviewer #2

In this manuscript Hang et al. introduce a theoretical model which accounts for the essential mechanical properties of cell membrane, cytoplasm and cytoskeleton. They demonstrate that their model can correctly reproduce not only the universal power-law characteristics of cell rheology but also the relationship between the power-law exponent and cell stiffness. In fact, the power-law exponent is found in this study to be in the range between 0.1 and 0.5, as observed in many experiments, and decreases linearly with the logarithm of cell stiffness. To the best of my knowledge, these results together can not be captured by any existing model of cell mechanics. The Authors also develop a self-similar hierarchical model that can reproduce the power-law dependences of the creep compliance with time and of the storage and loss moduli with frequency. Taken together, these method developments are noteworthy and will contribute significantly to the progress in cell mechanics.

The conclusions of this work are clearly supported by the presented results. I do not find any flaws in the data analysis, interpretation or conclusions. The methodology is sound. The only weakness of this manuscript is that there may be not enough detail provided in the methods for the work to be reproduced. Since the Authors use commercial finite-element software (Abaqus), they could simply deposit exemplary input/output files alongside with Supplementary Notes. It also would be a fine service to the community if the Authors could deposit the Python scripts used in this work to generate the cellular configurations and evaluate the finite-element model solutions.

Answer: We thank the reviewer for his/her positive recommendation of our paper. In response to this comment, we have added more details related to our modeling in the "Methods" section, and provided Python scripts along with the paper to ensure that the results can be reproduced by interested readers.

We have added the following statements in the "Methods" section (Page 17) and provided the Python scripts (see Supplementary Materials).

"For both step and cyclic loads, we simulated the viscoelastic cytoplasm and viscoelastic membrane by the Kelvin-Voigt model with the following constitutive relations:

$$\sigma_{\text{cyto}} = E_{\text{cyto}} \varepsilon_{\text{cyto}} + E_{\text{cyto}} \tau_{\text{cyto}} \dot{\varepsilon}_{\text{cyto}},$$

$$\sigma_{\text{mem}} = E_{\text{mem}} \varepsilon_{\text{mem}} + E_{\text{mem}} \tau_{\text{mem}} \dot{\varepsilon}_{\text{mem}},$$

where E_{cyto} and E_{mem} represent the elastic moduli of the cytoplasm and membrane, and τ_{cyto} and τ_{mem} the relaxation times of the cytoplasm and membrane, respectively. For microtubules (MTs) and microfilaments (MFs), we adopt linear elastic constitutive relations with corresponding elastic moduli E_{MT} and E_{MF} . The relevant parameters were taken as: $E_{\text{cyto}} = 100 \text{ Pa}$, $\tau_{\text{cyto}} = 300 \text{ s}$, $E_{\text{mem}} = 1000 \text{ Pa}$, $\tau_{\text{mem}} = 20 \text{ s}$, $E_{\text{MT}} = 1200 \text{ MPa}$, and $E_{\text{MF}} = 2400 \text{ MPa}$. A detailed description of the Kelvin-Voigt model can be found in Supplementary Note 1. The detailed geometric parameters of the cell structure can be seen in the “Model” section. All simulations were carried out by using the commercial finite element software Abaqus 6.13-1 and can be set automatically by running a python script (see Supplementary Materials) in Abaqus 6.13-1.”

Suggestion: *The predictions of the new model seem to be in quantitative agreement with a vast variety of experiments involving different cell types and states. I am wondering if it would be useful to present (in a new figure) the universal master curve of reference 14 (Annu. Rev. Mater. Res. 41, 75-97, 2011) and contrast it with the simulation results reported here.*

Answer: We thank the reviewer for this suggestion. As presented in Fig. 6 of Ref. 14, the power-law exponent can be collapsed into a universal master curve which decreases linearly with the cell stiffness in a semi-logarithmic coordinate. As predicted by Equation (7) in our manuscript, we show that when the cellular stiffness is not high, the creep compliance curves can intersect at a point (τ_0, j_0) , and the power-law exponent decrease linearly with the cell stiffness in a semi-logarithmic coordinate. This prediction is consistent with the universal master curve of Ref. 14. Moreover, when the cellular stiffness is high, the power-law exponent tends to be a constant, as shown in Fig. 5b in our manuscript and reported in experiments of Ref. 5. We have summarized many experimental data related to different cell types and cell states [Refs. (5, 11, 45, and 46)] and found that our predictions agree well with the experimental results (see Fig. R1).

In response to this comment, we have added the following statements on the relationship between the power-law exponent and cell stiffness (Page 14), a new Figure 6 (Page 15), and relevant references of experimental results.

“Here, we summarize existing experimental results^{5, 11, 45, 46} for different cell types and states, and plot the power-law exponent with respect to the cellular stiffness, as shown in Fig. 6. It is clearly seen that our predictions agree well with the experimental results and the cells become more solid-like as their stiffness increases. These results

confirm our predictions that for moderate cellular stiffness, the power-law exponent decrease linearly with the cell stiffness in a semi-logarithmic plot. Moreover, the power-law exponent of cells gradually converges to a certain threshold with increasing stiffness, which was not discussed in previous literature ¹⁴. These broad agreements between experimental findings and our predictions show the robustness of our self-similar hierarchical model in describing cell rheology.”

Fig. R1 | The relation between power-law exponent and normalized cellular stiffness. The power-law exponent α versus cellular stiffness of different cell types and states in experiments ^{5, 11, 45, 46} collapse onto a master curve. The stiffness of the cell corresponds to the inverse of the creep compliance $J(t)$ measured at time $t=1$ s or the storage modulus $G'(\omega)$ measured at an angular frequency $\omega=1$ Hz .

Reviewer #3

This manuscript described a theoretical framework to quantitatively understand the widely observe powerlaw rheology behavior of living cells. To my knowledge, this has not been achieved previously in a way as quantitatively as this manuscript. This model thus would be very valuable in understanding cells and their mechanical behaviors under different conditions. Thus I think this manuscript met the standard of Nature Communications. Nevertheless, I have some comments that should be addressed prior to publication.

Questions:

1. Power-law power is adjusted by the amount of MT as MT is an elastic component here. This suggests different power in experiments are from a different MT content which is not necessarily the case. I do understand the difficulty of the modeling that it is unnecessary to include more diverse components. But the authors should discuss this result, especially what it indicates to avoid confusion or misinterpretation.

Answer: We agree with the reviewer that some factors, such as the amount of MT and the viscoelasticity of cytoplasm, can result in different power-law exponents of cell rheology. As reported in many experiments (e.g., Refs. 10–14), the power-law exponents of cell rheology vary in the range of 0.1~0.5, depending on the cell types or cell states. By varying the amount of MT, we showed that the power-law exponent can be quantitatively tuned, which may explain why the power-law exponent differs for different cell types or states. In fact, other factors, such as the viscoelasticity of cytoplasm, the amount of microfilaments and intermediate filaments can also quantitatively regulate the power-law exponent.

In response to this comment, we have clarified some statements in the revised manuscript and added more discussion on the changes of the power-law exponent (Page 7).

“It can be seen that the increase in the amount of MTs can reduce the power-law exponent from 0.564 to 0.189.”

“In fact, changes in MT number and stiffness are among a number of factors that can alter the power-law exponent of cells. Similarly, changes in mechanical properties of other components of the cytoskeleton (MFs^{5, 6} and intermediate filaments^{36, 37}) or the cytoplasm^{38, 39} can also regulate the power-law exponent of cells. Therefore, it is possible that through re-configuring the network of the cytoskeleton or changing the mechanical properties of the cytoplasm, the power-law exponent can be quantitatively tuned in the range of 0.1 ~ 0.5, which may explain why the power-law exponent differs for different cell types or states (e.g., drug-induced)^{5, 7, 8}.”

2. Is ita on page 8 the same viscosity as the one used in Table 1, as they have the same symbol? If this is the background fluid, this typically is thought to be a few times more viscous than water according to the diffusion test, as it is an aqueous solution with proteins in it. The viscosity in Table 1 should be more like an effective viscosity of the entire cytoplasm. Otherwise, a timescale of 300 s won't make sense. Or this viscosity here is not the viscosity of the background fluid but rather an effective viscosity of the entire cytoplasm? This should be clarified.

Answer: The symbol η in the Kelvin-Voigt model (Table 1) and the self-similar hierarchical model (Figure 3 on Page 8 in the previous version; Page 9 in this revised version) has the same physical meaning and represents the effective viscosity of the entire cytoplasm. The cytoplasm is a crowded aqueous solution filled with ions and proteins. Hence, different cells exhibit different viscosities affected by the volume fraction of each component in the cytoplasm, as well as the interaction between the cytoplasm and the cytoskeleton. In this work, the effective viscosity of the cytoplasm is represented by η throughout the whole manuscript. To avoid confusion or misinterpretation, we have clarified this point in the revised manuscript.

In response to this comment, we have added the following statements in the revised manuscript (Page 4).

"The cytoplasm is a crowded aqueous solution filled with ions and proteins. Hence, different cells exhibit different viscosities, depending on the volume fraction of each component in the cytoplasm, as well as the interaction between the cytoplasm and the cytoskeleton. In this sense, the viscous coefficient η represents the effective viscosity of the entire cytoplasm."

3. Comparing Fig 4a (untreated) and 4d (cytoD treated), why are E2 and E3 change upon cytochalasin D treatment? CytoD dissolves F-actin but E2 is the stiffness of microtubules. Discussion regarding this would be helpful.

Answer: We thank the reviewer for pointing out this important issue. From a macroscopic perspective, the cell is treated as a 3-level self-similar hierarchical structure with E_1 , E_2 , and E_3 representing, respectively, the effective stiffness of the cytoplasm, MTs in the load direction, and the transverse expansion of the cytoskeleton and the cytoplasm, and η representing the effective viscosity of the entire cytoplasm. Since the drug cytochalasin D can dissolve actin filaments, there will be a significant reduction in the effective stiffness of the cytoskeletal network in both loading and transverse directions, i.e., a reduction in both E_2 and E_3 (Fig. 4d). In addition, the results of Fig. 4b (Histamine treated) and Fig. 4c (DBcAMP treated) are also discussed in the revised manuscript.

In response to this comment, we have added some statements (Page 9) and more discussion on the effects of different drugs on the changes of mechanical properties of cells (Page 11) in the revised manuscript.

"In this way, from a macroscopic perspective, the cell is treated as a 3-level self-similar hierarchical structure with E_1 , E_2 , and E_3 representing, respectively, the effective stiffness of the cytoplasm, MTs in the load direction, and the transverse expansion of the cytoskeleton and the cytoplasm, and η representing the effective viscosity of the entire cytoplasm."

"The drug Histamine ⁴³ can enhance the permeability of cells, which reduces the cytoplasmic stiffness E_1 (see Fig. 4(b)). This drug also promotes cell contraction that can increase the stiffness of the cytoskeletal network (E_2 and E_3). For cells treated with DBcAMP, the contraction of cells is inhibited ^{8, 44}, which reduces the stiffness of the cytoskeletal network (E_2 and E_3), as shown in Fig. 4(c). When the cells are treated with cytochalasin D ⁸, the cytoskeleton is dissolved, resulting in a reduction in stiffness (E_2 and E_3) of the cytoskeletal network (see Fig. 4(d))."

4. While this model successfully describes the power-law rheology measured on the level of the whole cell and the cortex, the model itself doesn't separate the thick F-actin rich cortex and the more diluted internal cytoplasm. Nevertheless, the cytoplasm also shows the classical power-law rheology as shown by many groups (for example <https://doi.org/10.1016/j.cell.2014.06.051>), with a power ranging from 0.1 to 0.5 as well. It seems this model can also do a great job in describing the power-law rheology

observed in the cytoplasm alone. As the cytoplasm is known to be much softer than the whole cell or the cell cortex, this distinction would be important to acknowledge. It would be great if some discussion can be added.

Answer: We thank the reviewer for pointing out this extension of our model. Experiments showed that the elastic modulus of the cytoplasm also follows a power-law dependence on loading frequency: $G' \sim \omega^\beta$ with exponent $\beta = 0.15$. In our model, the structural details of the cytoplasm are ignored, since it is much softer than the cytoskeleton. Thus, the whole cytoplasm is considered as a viscoelastic medium, i.e., the 1st level hierarchy of the model. When one studies the rheological response of the local region of the cytoplasm, the structural details of the cytoplasm must be considered, and the self-similar hierarchical model can then be used to study the rheology of the cytoplasm. The interstitial fluid inside the cytoplasm (containing water, ions and small proteins) can be considered as the 1st level hierarchy, the large proteins in the cytoplasm as the 2nd level hierarchy, and the interaction between the large proteins as the 3rd level hierarchy. In this sense, the present model can be extended to investigate the dynamical mechanical response of the cytoplasm.

In response to this comment, we have added some discussion on the extension of our model on the cytoplasm (Pages 11 and 12) and also added relevant references.

“In addition, the self-similar hierarchical model can also be used to study the power-law rheology observed in the cytoplasm whose storage modulus follows a similar power-law form $G' \sim \omega^\beta$ with $\beta = 0.15$ ³⁸. When using this model to investigate the rheological response of the cytoplasm, the structural details of the cytoplasm should be considered. The interstitial fluid inside the cytoplasm (containing water, ions and small proteins) can be treated as the 1st level hierarchy, the large scale proteins in the cytoplasm as the 2nd level hierarchy, and the interactions between the proteins as the 3rd level hierarchy. In this sense, the present model can be extended to investigate the dynamical mechanical response of the cytoplasm.”

“With the self-similar hierarchical model, one can describe, explain, and predict the rheological behavior of living cells with different types or states, as well as the viscoelastic cytoplasm.”

5. *On page 7, it is stated that MTs and microfilaments cross and connect with each other. They indeed interpenetrate with each other but not necessarily connect to each other.*

Answer: In response to this comment, we have revised the relevant descriptions in the manuscript (Page 8).

“In cells, abundant MTs and microfilaments **interpenetrate** with each other to form a three-dimensional cytoskeleton network bathed in the cytoplasm⁴⁰⁻⁴² **composed of water, solutes, and small molecules.**”

6. *It is known that the cell cytoskeleton is composed of F-actin, MTs and also*

intermediate filaments. Recently, the role of cytoskeletal intermediate filaments in cell mechanics has become more clear. For example, <https://doi.org/10.1073/pnas.1903890116>. Given each component may have a different mechanical contribution to the overall mechanics, can the authors add discussion on this?

Answer: We thank the reviewer for pointing out the role of intermediate filaments (IFs) in cell mechanics. Indeed, as with microtubules (MTs) and microfilaments (MFs), IFs also play an important role in cell mechanics, as reported in recent experiments (Ref. 36). When studying the cell's creep response under small deformations, the effect of IFs can be ignored, since they contribute little to the cortical stiffness in this case (Ref. 47). As pointed by the reviewer, vimentin IFs have a critical mechanical contribution to the overall cell mechanics especially at large deformations, substantially enhancing the strength, stretchability, resilience, and toughness of cells (Ref. 36). We showed that by treating IFs and MFs as strings in a prismatic tensegrity structure (Supplementary Note 4), the cells can exhibit significant strain-stiffening behavior as found in many experiments (Refs. (15-17, and 36)).

In response to this comment, we have added some discussion on the effect of IFs (Pages 15 and 16) and Supplementary Note 4 in the revised manuscript.

“When studying the creep response of cells under small deformations, we have ignored the effect of intermediate filaments (IFs), since they contribute little to the cortical stiffness in this case⁴⁷. Very recently, Hu et al. studied the effect of IFs on the mechanical properties of cells, and showed that under large deformations, the IF network behaves as a strain-stiffening hyperelastic network that substantially enhances the strength, stretchability, resilience, and toughness of cells³⁶. Supplementary Note 4 shows that by treating IFs and MFs as strings in a prismatic tensegrity structure, the cells can exhibit the remarkable strain-stiffening behavior found in experiments^{15-17, 36}, while holding the rheological characteristics. In addition, IFs play an important role in the mechanics of epithelial monolayers^{37,48}, which can also be studied by our model. This suggests a strong potential of self-similar hierarchical models for investigating the mechanics of natural biological materials.”

REVIEWERS' COMMENTS

Reviewer #2 (Remarks to the Author):

The Authors have satisfactorily responded to my comments and suggestions, and revised their manuscript accordingly. I have no further comments and recommend publication of the manuscript in Nature Communications.

Reviewer #3 (Remarks to the Author):

The authors addressed all my questions and I don't have any further comments on this work. It reads very well!

Response to Reviewers

The authors wish to thank the reviewers for their very helpful comments and suggestions.

Reviewer #2

The Authors have satisfactory responded to my comments and suggestions, and revised their manuscript accordingly. I have no further comments and recommend publication of the manuscript in Nature Communications.

Answer: We thank the reviewer for his/her comments and suggestions that have improved the manuscript.

Reviewer #3

The authors addressed all my questions and I don't have any further comments on this work. It reads very well!

Answer: We thank the reviewer for his/her comments and suggestions that have improved the manuscript.